

# Opioid distribution trends (2006–2017) in the US Territories

Fedor F. Cabrera[1], Erik R. Gamarra[1], Tiffany E. Garcia[1], Ashanti D. Littlejohn[1], Poul A. Chinga[2], Luis D. Pinentel-Morillo[3], Jorge R. Tirado[4], Daniel Y. Chung[1], Leana J. Pande[5], Kenneth L. McCall[6], Stephanie D. Nichols[7] and Brian J. Piper[1,8]

[1] Department of Medical Education, Geisinger Commonwealth School of Medicine, Scranton, PA, United States of America
[2] Department of Biology, University of Scranton, Scranton, PA, United States of America
[3] Department of Biology, Pennsylvania State University, State College, PA, United States of America
[4] Department of Biology, Elizabethtown College, Elizabethtown College, PA, United States of America
[5] Department of Biology, Wilkes University, Wilkes-Barre, PA, United States of America
[6] Department of Pharmacy Practice, University of New England, Portland, ME, United States of America
[7] Department of Pharmacy Practice, Husson University School of Pharmacy, Bangor, ME, United States of America
[8] Center for Pharmacy Innovation and Outcomes, Geisinger Precision Health Center, Forty Fort, PA, United States of America

Corresponding author
Brian J. Piper,
bpiper@som.geisinger.edu,
psy391@gmail.com

## ABSTRACT

**Background**. The US mainland is experiencing an epidemic of opioid overdoses. Unfortunately, the US Territories (Guam, Puerto Rico, and the Virgin Islands) have often been overlooked in opioid pharmacoepidemiology research. This study examined common prescription opioids over the last decade.

**Methods**. The United States Drug Enforcement Administration's Automation of Reports and Consolidated Orders System (ARCOS) was used to report on ten medical opioids: buprenorphine, codeine, fentanyl, hydrocodone, hydromorphone, meperidine, methadone, morphine, oxycodone, and oxymorphone, by weight from 2006 to 2017. Florida and Hawaii were selected as comparison areas.

**Results**. Puerto Rico had the greatest Territorial oral morphine mg equivalent (MME) per capita (421.5) which was significantly higher ($p < .005$) than the Virgin Islands (139.2) and Guam (118.9) but significantly lower than that of Hawaii (794.6) or Florida (1,509.8). Methadone was the largest opioid by MMEs in 2017 in most municipalities, accounting for 41.1% of the total in the Virgin Islands, 37.9% in Florida, 36.6% in Hawaii but 80.8% in Puerto Rico. Puerto Rico and Florida showed pronounced differences in the distribution patterns by pharmacies, hospitals, and narcotic treatment programs for opioids.

**Conclusions**. Continued monitoring of the US Territories is needed to provide a balance between appropriate access to these important agents for cancer related and acute pain while also minimizing diversion and avoiding the opioid epidemic which has adversely impacted the US mainland.

## INTRODUCTION

Increasing rates of opioid prescriptions and overdoses in the past decade and a half have led epidemiologists and lawmakers to refer to the current situation as an opioid epidemic. Almost two-hundred people die each day in the United States (US) from overdoses (*Sanger-Katz, 2018*).

Lawmakers and healthcare administrators have developed several strategies to address opioid misuse. These include removing pain as the fifth vital sign and the implementation of prescription take back and drug disposal programs. Increased efforts have been made to change the focus of patient care surveys and insurance reimbursement towards how pain is addressed, as opposed to encouraging prescription of opioids. Prescription Drug Monitoring Programs (PDMP) are integral in tracking prescription medications (*Patrick et al., 2016*). PDMPs programs have been incorporated since 1939 in California and subsequently across the United States. Currently, all states (except Missouri), the District of Columbia, and Guam have operational PDMPs (*Prescription Drug Monitoring Program Training and Technical Assistance Center, 2018*; *President's Commission on Combatting Drug Addiction and the Opioid Crisis, 2017*). The rates of opioid prescription vary by geographic location and was possibly affected by the presence of an active PDMP (*Curtis, Stoddard & Radeva, 2006*).

There are pronounced disparities between the United States and the Territories. While Medicaid federal spending is matched with the state spending level, to no limit, there is a cap on the amount of money provided to the Territories (*Nunez-Smith et al., 2011*). The impact of the Medicaid spending cap is overt, denying coverage of mandatory and optional services needed by the patient population of the Territories. These healthcare services include preventative screenings, home health nursing, rehabilitation centers, and hospice care, and others (*Gutierrez, 2011*). Importantly, the Territories are under the jurisdiction of the Food and Drug Administration, Drug Enforcement Administration and the Centers for Disease Control and Prevention. Puerto Rico, the most populated of the Territories, was included in an opioid pharmacoepidemiology report although this investigation did not include opioids provided by narcotic treatment programs (NTP) (*Félix, Mack & Jones, 2016*). Notably, when Puerto Ricans from the US mainland were compared to Puerto Ricans on the islands, there was a higher prevalence of substance-use-related consequences, such as HIV and hepatitis among the island population (*Hautala et al., 2017*). Although the Virgin Islands and American Samoa do not have PDMPs, Guam has had an active program since 2013. Puerto Rico is making strides towards this goal and passed PDMP legislation in 2016 and the program became operational in June, 2018.

Overall, given the few studies that have focused on the US Territories specifically and the use and misuse of prescription of opioids (*Battiste, Ryan & Engerman, 2017*; *Félix, Mack & Jones, 2016*; *Hautala et al., 2017*), it is unclear whether these locations have been experiencing the same trends in opioid prescriptions as the mainland US (*Piper et al., 2018*). The present study aimed to investigate the temporal pattern in opioid distribution in the US Territories of Guam, Puerto Rico, the Virgin Islands, and American Samoa. Florida and Hawaii were used in comparison due to their geographic and demographic similarities to
the Caribbean and Pacific islands, respectively (Table S1). We also examined the various distribution sources (i.e., pharmacy, hospital, NTP) across locations to determine what facilities make opioids available. The results also seek to inform whether prescription opioids are being distributed more on an outpatient or inpatient basis or for pain versus addiction.

## MATERIALS AND METHODS

### Data source

The Automation of Reports and Consolidated Orders System (ARCOS) was used to collect information on the distribution of prescription opioids and their respective distributors from 2006 to 2017 (*US Department of Justice, 2018*). ARCOS is a comprehensive drug reporting system that is administered by the US Drug Enforcement Administration to track the distribution of controlled substances. Manufacturers and distributors are required to report controlled substances transactions including inventory, point of sale, and distribution of substances at the dispenser/retail level. ARCOS tallies the cumulative and quarterly sale of controlled substances in grams and reflects the distribution to pharmacies, hospitals, practitioners, mid-level practitioners, teaching institutions, and Narcotic Treatment Programs (NTP). Nine opioids, oxycodone, oxymorphone, hydrocodone, morphine, methadone, meperidine, fentanyl, codeine, and hydromorphone, were selected based on reports of their being commonly abused (*Drug Enforcement Administration, 2016*; *Index Mundi, 2018*; *National Institute of Drug Abuse, 2018*; Table S2). Buprenorphine was chosen due to its increasing use (*Piper et al., 2018*) in treating an opioid use disorder, although it is also employed for pain. Procedures were approved by the IRB of the University of New England (#20180410-009).

### Statistical analysis

Four analyses were completed: (1) the total oral morphine milligram equivalent (MME) was calculated and converted to kg for all ten opioids and expressed per year (2006 to 2017) for each territory; (2) percent change for each opioid relative to 2006; (3) the MME per capita with US Census, American Community Survey, and other sources (*American Fact Finder, 2018*) used to determine population (Fig. S1); (4) total distribution to pharmacies, hospitals, practitioners, and NTPs expressed as a percentage of the total. The oral MME was calculated to correct for the relative potency of each opioid relative to morphine. The conversions were completed using the following factors: buprenorphine (10), codeine (0.15), fentanyl base (75), hydrocodone (1), hydromorphone (4), meperidine (0.1), methadone (12 from NTP and 8 from other sources), morphine (1), oxycodone (1.5), and oxymorphone (3) (*Piper et al., 2018*). A paired $t$-test compared the MME per capita by location for each year (2006–2017). Data analysis and figures were completed with Systat, version 13.1 and GraphPad Prism, version 7.04.

## RESULTS

Figure 1A shows the change in opioid distribution from 2006 to 2017 after conversion to oral MME. Municipalities generally showed an inverted-U pattern with initial increases in

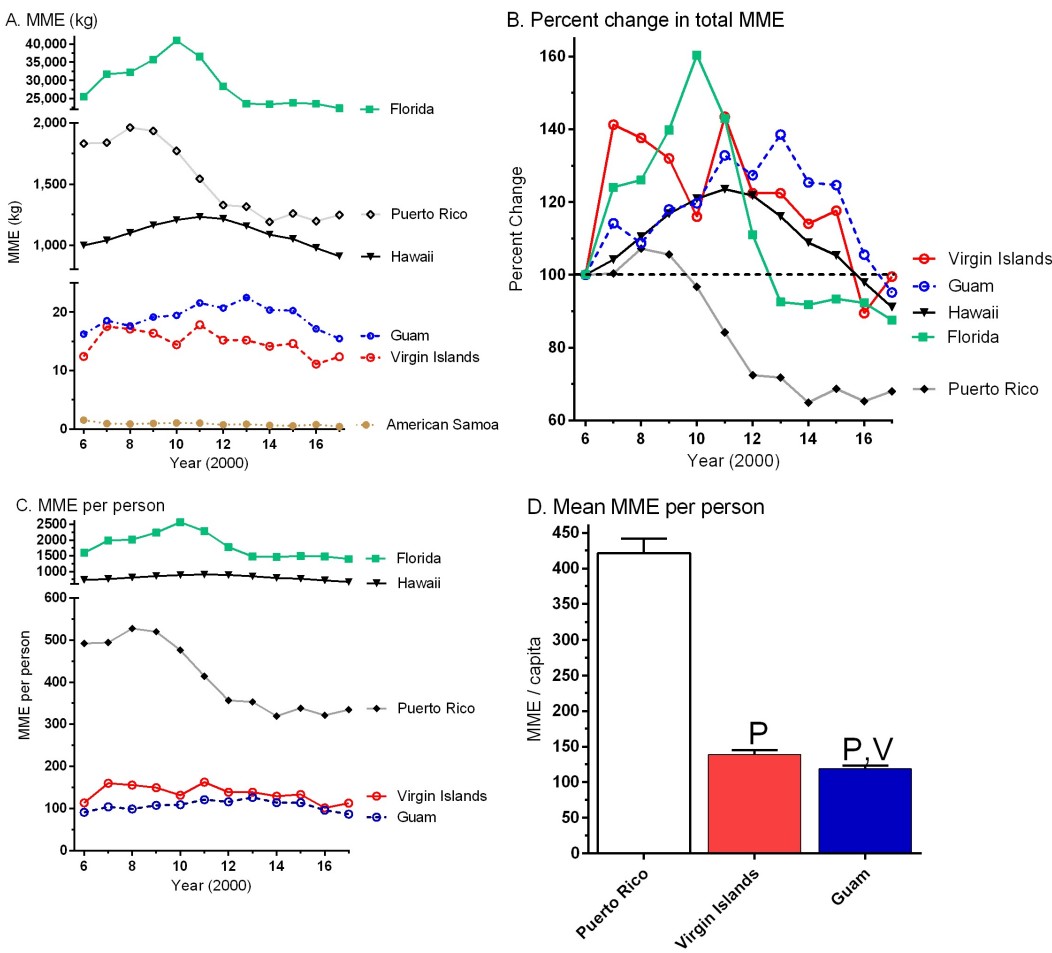

**Figure 1** **Total distribution of ten opioids from 2006 to 2017.** Total distribution of ten opioids (buprenorphine, codeine, fentanyl, hydrocodone, hydromorphone, meperidine, methadone, morphine, oxycodone, and oxymorphone), uncorrected for population (A), expressed as a percent change in weight relative to 2006 (B), and corrected for population (C) in morphine mg equivalents (MME) between 2006 and 2017 by location as reported to the US Drug Enforcement Administration's Automation of Reports and Consolidated Ordering System. (D) Mean (± SEM) MME/capita in the US Territories. [P]$p < .0005$ versus Puerto Rico, [V]$p < .0005$ versus the US Virgin Islands.

the kg of opioids distributed followed by decreases over time. Puerto Rico peaked in 2008, Florida in 2010, Hawaii in 2011, and Guam in 2013. There were MME increase of +4.2% in Puerto Rico, +11.2% in the Virgin Islands versus decreases of −9.9% in Guam, −5.1% in Florida, and −6.9% in Hawaii from 2016 to 2017. Figure 1B, with data expressed as a percentage of 2006, more clearly illustrates the temporal pattern over the last decade with peaks in 2007 (+41.2%), 2008 (+7.2%), and 2013 (+38.5%) for the Virgin Islands, Puerto Rico, and Guam, respectively. Figure 1C shows the MME when corrected for population. The MME per person from 2006 to 2017 in Florida (1,509.7 ± 107.5) was almost two-fold higher than Hawaii (794.6 ± 23.5, $t(11) = 7.94$, $p < .0005$). Figure 1D illustrates that the MME per person in Puerto Rico (421.5 ± 20.3) was three-fold, and significantly elevated

**Table 1  Percent Change in opioids by weight (g) from 2006 to 2017 as reported by the US Drug Enforcement Administration's Automation of Reports and Consolidated Ordering System.**

| Opioid | Florida | Puerto Rico | Hawaii | Guam | US Virgin Islands | United States & Territories |
|---|---|---|---|---|---|---|
| buprenorphine | +955.69%[a] | +1,654.05%[a] | +750.67%[a] | −57.48%[b] | +73.62%[a] | +1,058.16%[a] |
| codeine | −24.90%[b] | +8.58% | −39.16%[b] | −54.59%[b] | −33.85%[b] | −13.91%[b] |
| fentanyl | −12.48%[b] | +23.77%[a] | −30.13%[b] | −0.89% | −39.07%[b] | −24.28%[b] |
| hydrocodone | −37.53%[b] | −87.06%[b] | −9.17% | −37.48%[b] | −29.12%[b] | −9.13% |
| hydromorphone | +135.92%[a] | +140.55%[a] | +16.40%[a] | +104.91%[a] | +120.62%[a] | +55.72%[a] |
| meperidine | −87.86%[b] | −63.58%[b] | −86.75%[b] | −71.71%[b] | −82.78%[b] | −83.21%[b] |
| methadone | −41.15%[b] | −38.80%[b] | −35.44%[b] | −26.16%[b] | −7.86% | +5.18% |
| morphine | +26.43%[a] | −23.80%[b] | −29.70%[b] | −14.41%[b] | −6.23% | −5.16% |
| oxycodone | +3.67% | −27.68%[b] | +17.51%[a] | +41.59%[a] | +85.11%[a] | −31.92%[b] |

Notes.
[a]Positive change by >10%.
[b]Negative change by >10%.

**Table 2  Opioids, ranked by percent of total morphine mg equivalent of ten opioids as reported by the US Drug Enforcement Administration's Automation of Reports and Consolidated Ordering System in 2017.**

| Rank | Florida | Puerto Rico | Hawaii | Guam | US Virgin Islands | American Samoa | United States & Territories |
|---|---|---|---|---|---|---|---|
| 1 | meth 37.9% | meth 80.7% | meth 36.6% | oxy 33.2% | meth 41.1% | codeine 47.8% | meth 46.9% |
| 2 | oxy 27.5% | bup 8.0% | oxy 28.6% | mor 25.8% | oxy 25.5% | oxy 20.6% | oxy 20.9% |
| 3 | bup 8.4% | oxy 6.1% | mor 9.6% | fentanyl 20.1% | hyd 9.4% | mor 20.5% | bup 9.6% |
| 4 | fentanyl 7.8% | fentanyl 3.5% | bup 8.7% | meth 8.9% | fentanyl 8.5% | fentanyl 8.2% | hyd 7.7% |
| 5 | mor 7.0% | mor 0.7% | hyd 7.7% | codeine 4.6% | bup 5.1% | hyd 1.9% | fentanyl 6.9% |

Notes.
bup, buprenorphine; hyd, hydrocodone; meth, methadone; mor, morphine; oxy, oxycodone.

relative to the Virgin Islands (139.2 ± 5.8). Further, the Virgin Islands had a higher MME than Guam (118.9 ± 4.1, $t(11) = 4.13$, $p < .005$) and American Samoa (15.8 ± 1.4, $t(11) = 20.36$, $p < .0005$).

Table 1 expresses the percent change in each opioid in the three US Territories, Hawaii, and Florida from 2006 to 2017. Meperidine showed large reductions in all municipalities. Methadone exhibited uniform decreases across locations. Morphine, fentanyl, and codeine quantities declined and buprenorphine and oxycodone increased in four of the five assessed areas.

Table 2 shows that methadone was the most prevalent of the ten opioids by MME in the majority of areas. Buprenorphine was in the top five in four of the six municipalities. Oxycodone was consistently among the top three. Nationally, methadone from NTP (40.0%) and buprenorphine accounted for almost half (49.6%) of the total opioid MME (Fig. S2).

The source of opioid distribution, by percent of the total MME, varied widely across locations. NTPs were responsible for none (0.0%) in Guam versus over four-fifths (80.5%) in Puerto Rico. Pharmacies were responsible for the preponderance of opioid distribution
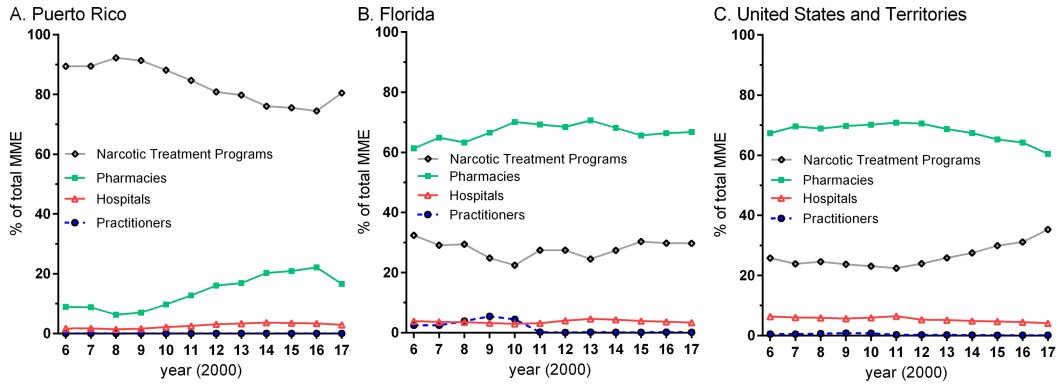

**Figure 2** **Oral morphine mg equivalent (MME) for ten opioids.** Percent of the total oral morphine mg equivalent (MME) for ten opioids (buprenorphine, codeine, fentanyl base, hydrocodone, hydromorphone, meperidine, methadone, morphine, oxycodone, and oxymorphone) by business activity as reported by the US Drug Enforcement Administration's Automation of Reports and Consolidated Ordering System for Puerto Rico (A), Florida (B), and the United States and Territories (C).

(>60%) except in Puerto Rico (16.6%) in 2017. NTPs were responsible for over three-quarters of prescription opioids in Puerto Rico (Fig. 2A) relative to less than one-third in Florida. There were dynamic changes in pharmacy's MME while hospitals showed smaller fluctuations. Practitioners exceeded hospitals from 2008 to 2010 in Florida (Fig. 2B). The entire US, including the Territories (Fig. 2C) generally reported a similar pattern as Florida (Fig. S3).

## DISCUSSION

This study identified dynamic changes in prescription opioids from 2006 to 2017 in the US Territories of Guam, Puerto Rico, American Samoa and the Virgin Islands and the quantities distributed by business activity relative to Florida and Hawaii. The total MME of ten opioids (Fig. 1) showed an inverted-U pattern over the last dozen years with elevations followed by declines. The total MME per capita was calculated for each location and Puerto Rico's was only one-quarter that of Florida. American Samoa, Guam, and the Virgin Islands were significantly lower than Puerto Rico which is congruent with ARCOS data that focused more narrowly on 2016 (*Piper et al., 2018*).

It is reasonable to speculate that the recent de-escalation of opioid distribution resulted from heightened awareness of the opioid addiction crisis the country has been experiencing over the past decade (*President's Commission on Combatting Drug Addiction and the Opioid Crisis, 2017*). This decline can also be seen in Table 1 which examines the percentage changes between 2006 and 2017 where some opioids saw a decrease in their distribution. This was particularly true for meperidine. Concern about the metabolite normeperidine which has a long-half life, is biologically active, and can cause altered mental status, psychosis, myoclonus, and seizures, have prompted several organizations to discourage use of meperidine (*Dobbins, 2010*). In contrast, buprenorphine showed pronounced elevations which likely results from increasing use to treat an opioid use disorder. Prescription opioid

use reflects a confluence of factors including restrictions in supply, changing demands due to prescribing guidelines and patient expectations, and pharmacoeconomic factors which reflect patent expirations and finite health care resources. Morphine, fentanyl, and methadone are on the United Nations List of Essential Medications and were prominent in the Territories and States.

One of the most striking findings in this report was that Puerto Rico's most abundant prescription opioid by morphine equivalents from 2006 to 2017 was methadone. The preponderance of methadone in Puerto Rico indicates that the residents in this location are accessing the six NTPs. Conversely, opioid use for pain is modest relative to the mainland US (*Piper et al., 2018*), resulting in a heightened proportion from NTP. These findings should be interpreted with caution and not be used to stigmatize Puerto Rico. Guam and American Samoa do not have a NTP. Methadone is an evidence based treatment for an opioid use disorder (*Faggiano et al., 2003*). Nationally, methadone accounted for 46.9% of the MME which was twice as much as the next largest opioid, oxycodone and six times more than hydrocodone. A recent review indicated that methadone represented only 1% of all opioid prescriptions (*Manchikanti et al., 2018*). An underappreciation for the importance of methadone on a population level results from a variety of factors including: (1) a well-intentioned regulation (42 CFR Part 2) that prevents methadone as part of opioid use disorder treatment from being entered into PDMPs and other barriers to access of this information; (2) use of outdated and low MME conversion factors in pharmacoepidemiology research (*Guy et al., 2017*); and (3) the separate administration system of NTPs in the US results in many fewer health care providers, relative to buprenorphine, having direct experience with methadone. Others have called for the reconsideration of 42 CFR Part 2 (*President's Commission on Combatting Drug Addiction and the Opioid Crisis, 2017*) as this regulation is an obstacle to physicians and pharmacists knowing which opioid pharmacotherapies, and the dose, their patients may be receiving. Future legal innovations in the Territories could result in more complete methadone reporting than currently occurs in the states.

Puerto Rico has seen implementation of policies and programs for mental health and substance abuse (*Leff et al., 2017*). Compared to the other locations, methadone was not as prevalent in Guam where oxycodone, morphine, and fentanyl were more common. The data appears to support earlier work (*Curtis, Stoddard & Radeva, 2006*; *Félix, Mack & Jones, 2016*; *Piper et al., 2018*) in that the differences in opioid distribution varied by location. Although the Virgin Islands and Guam, having the smaller populations compared to Puerto Rico, were found to have similar abundance of opioid distribution, these locations are obviously not located near each other geographically. However, the Virgin Islands does have a notable distribution of opioids from NTPs indicating increased efforts towards addressing opioid use disorders.

It is noteworthy that Florida showed more dynamic changes in total opioid volumes over the past decade than Hawaii. Florida had 250 active pill mills and a law to close them was enacted in 2011 (*Chang et al., 2016*). One-twentieth of prescribers were responsible for two-thirds of the volume of opioids in Florida (*Chang et al., 2016*). Florida also led the US in direct dispensing of controlled substances by physicians but this dubious practice

underwent a pronounced decline since 2010 (*Hansen & Netherland, 2016*; *Mack, Jones & McClure, 2018*).

Ethnicity is a key consideration in opioid pharmacoepidemiology. Whites that visit emergency rooms in the mainland US are more likely to be prescribed opioids, particularly morphine and hydromorphone, than non-whites (*Pletcher et al., 2008*). Nationally representative surveys have determined that non-Hispanic whites both used heroin and misused nonmedical prescription opioids prior to initiating heroin more frequently than non-whites (*Martins et al., 2017*). Further study is necessary to determine if non-white ethnicity, or associated variables, are similarly protective against opioid misuse in the US Territories or if the dynamic of being in a "majority–minority" community (Table S1) impacts these associations.

Our research on the US Territories had a few limitations and caveats. Because we focused on the prescription opioids reported to be the most highly used and misused, Schedule IV agents like tramadol, which is also not monitored by ARCOS, was not reported. Tapentadol did become available until 2008 and has an unusual mechanism of action (*Kögel et al., 2011*). As a result, the MME reported in Fig. 1 may be an underestimate. The data obtained from ARCOS includes opioids which are reported to the DEA. There is a probability that some opioids may be obtained from sources not recorded in this dataset (e.g., from mail-order pharmacies outside of the investigated municipalities) (*Khandelwal et al., 2012*). Figure 1C and 1D should be interpreted based on the recognition that the MME per capita was based on population estimates for non-Census years which may overestimate the denominator for 2017 in Puerto Rico following hurricane Maria due to migration (*Kishore et al., 2018*). ARCOS also does not provide patient level data or the indications the opioids were being distributed which should be the topic of further investigation. Further public policy studies to further understand the origins of the changes observed including the pronounced fiscal constraints in health care spending in Puerto Rico are needed.

## CONCLUSION

In conclusion, this study characterized trends in opioid distribution in the US Territories of Guam, Puerto Rico, and the Virgin Islands. The distribution of prescription opioids in US Territories is different than the US mainland in, scale but less so for the temporal pattern. Overall, the US Territories had significantly lower distribution of opioids than Florida and Hawaii. Among the Territories, Puerto Rico had the highest average MME per capita with the preponderance of its distribution coming from NTPs. These differences between Territories and States regarding opioid use may be related to cultural and socioeconomic factors, healthcare disparities, public health policies as well as illicit drug availability. Given the large impact and attention on the opioid epidemic in the mainland US, it is also likely these trends are a secondary effect for the Territories. However, given the lack of active PDMPs, with the exception of Guam, there should continue to be improvements in surveillance of controlled substances in these locations. The US Territories have a population primarily consisting of racial minorities and therefore typically underserved and underrepresented in medicine. The opioid epidemic has had a greater impact on

White populations (*Hansen & Netherland, 2016*) and has been described as at least partially iatrogenic (*Piper et al., 2018*). Ironically, health care disparities could, in this very narrow context, protect the US Territories from an escalation from prescription opioids to heroin and illicit fentanyl. Further research needs to identify the specific sociocultural factors impacting these Territories and how they are interrelated with prescription opioid use and abuse. Although additional data regarding emergency room visits and overdoses are needed from the US Territories, Florida and other states might benefit from reflecting on whether having extremely low prescription opioid use has had population level benefits for the US Territories.

## ACKNOWLEDGEMENTS

Thanks to pharmacy faculty at the University of Puerto Rico and Olapeju M. Simoyan, MD for feedback about this project. The first four authors are listed in alphabetical order.

### Funding

Felix F. Cabrera, Erik R. Gamarra, Tiffany E. Garcia, Ashanti D. Littlejohn, Poul A. Chinga, Luis D. Pimental-Morillo, and Jorge R. Tirardo were supported by the Center of Excellence, Health Resources and Services Administration (D34HP31025) and Brian J. Piper was a Fahs-Beck Fellow. Software was provided by the GCSoM Summer Research Immersion Program, Husson University School of Pharmacy, and NIEHS (T32 ES007060-31A1). The funders had no role in study design, data collection and analysis, decision to publish, or preparation of the manuscript.

### Grant Disclosures

The following grant information was disclosed by the authors:
Center of Excellence, Health Resources and Services Administration: D34HP31025.
GCSoM Summer Research Immersion Program, Husson University School of Pharmacy.
NIEHS: T32 ES007060-31A1.

### Competing Interests

The authors declare there are no competing interests.

### Author Contributions

- Fedor F. Cabrera, Erik R. Gamarra, Tiffany E. Garcia and Ashanti D. Littlejohn conceived and designed the experiments, performed the experiments, analyzed the data, prepared figures and/or tables, authored or reviewed drafts of the paper, approved the final draft.
- Poul A. Chinga performed the experiments, authored or reviewed drafts of the paper, approved the final draft.
- Luis D. Pinentel-Morillo, Jorge R. Tirado and Leana J. Pande performed the experiments, approved the final draft.
- Daniel Y. Chung performed the experiments, prepared figures and/or tables, approved the final draft.

- Kenneth L. McCall and Stephanie D. Nichols conceived and designed the experiments, authored or reviewed drafts of the paper, approved the final draft.
- Brian J. Piper conceived and designed the experiments, analyzed the data, prepared figures and/or tables, authored or reviewed drafts of the paper, approved the final draft.

### Ethics

The following information was supplied relating to ethical approvals (i.e., approving body and any reference numbers):

Procedures were approved by the IRB of the University of New England (#20180410-009).

### Data Availability

The data is available at: https://www.deadiversion.usdoj.gov/arcos/retail_drug_summary/index.html.

Interested parties should download the data for each year of interest.

### Supplemental Information

Supplemental information for this article can be found online at http://dx.doi.org/10.7717/peerj.6272#supplemental-information.

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
