# Peer review of "Opioid distribution trends (2006–2017) in the US Territories"

_PeerJ, doi:10.7717/peerj.6272_

## Round 0.1 · original submission · Major Revisions

This paper is a relatively straightforward reporting of opioid use data in understudied areas of the US and dependencies. As it stands it serves to aggregate data, but little else. The reviewers indicate several areas for potential enrichment, in particular inclusion of data about opioid poisonings and about the number of individuals prescribed. It would also be interesting to know if there was any data about non-prescription opioid use in the areas under study. Finally, I think it would also be quite useful to know how the territories compared with the US as a whole, in addition to relevant geographically/demographically equivalent states. The authors state the territories are largely populated by minority groups, this is perhaps true when compared with the whole of the USA, but presumably local ethnicities are not minority groups in these locations, which gives under-representation/underserving a different dynamic than speaking about a large mainland city. Comparing to the US as a whole would also mitigate against the fact that Florida used to be a source of opioids for many people in addition to state residents – a key feature of these Territories (and Hawaii) is geographical isolation, and so the assumption that most opioids are being used by locals is reasonable. For Florida it is not. Provision of this data or these comparators is an opportunity and a request, not a requirement.

There are several things that the authors must to address in a revised paper. I share Reviewer’s concerns about the use of a paired t-test to compare MME per capita by location; paired t-tests are usually reserved for repeated measures from the same population, not measurement of the same thing in different populations. This should be redone. It is also unclear to me whether route of administration has been considered in MME calculations – in other words, why not use oral morphine equivalents. Other comments from the reviewers should be considered (i.e., the multiple uses of buprenorphine, possibility of changes in population of the time of the study).

The Results section restates many numbers from the Tables, which should be avoided. Comments about changes from 2016 to 2017 are really meaningless unless compared to changes in previous years. Even though the Results section is short as is, it could benefit from more focus and purposeful presentation of the data.

Reviewer 1 ·

Basic reporting

No comment

Experimental design

not applicable

Validity of the findings

See below under general comments

Additional comments

General comments
This article is basically reporting bulk distribution of opioids in 3 US territories and 2 US states – corrected with morphine equivalents and then for population. It doesn’t really do much more than describe these data (are these data publically available anyway?). It isn’t able to say whether changes are due to increased/decreased doses or more/less people? Have recent decreases due to reduced prescribing led to more illicit opioids? Can any changes be linked to any particular events ?
The key to making this paper interesting would be to have some outcome data such as overdose/poisoning deaths in each territory. At the moment it’s a small amount of data and a lot of speculation. If they can’t add more interesting data it would be much better as a brief report with very little introduction or discussion. The discussion is way too long for such a thin paper.
Specific points
The exclusion of American Samoa because of low volumes makes no sense.
Buprenorphine is used as an analgesic as well as for opioid dependence.
Opioids are usually corrected to oral morphine equivalents (ie conversion rates for IV, dermal and oral preps are different) – it is unclear if that has been done in these MME.
It would make more sense to report pop adjusted OTP and analgesic opioids separately – the % of total analysis (Fig 2) is less relevant than the actual rates of each.

·

Basic reporting

The language used is mostly clear and unambiguous. A couple of recommendations are suggested below to help increase clarity to readers.
Line 181-182: “Similarly, hospitals distributed 26.6% in Guam, …”. To increase clarify it would be better to say distributed 26.6% of opioids in Guam,….
Figure 1: Caption of Figure 1 needs to be rewritten to increase clarity. It’s missing (D).
Good background provided with sufficient reference.
Line 97-98: According to table provided US Virgin Islands is populated by mainly Black/African American which is different from comparison States. This need to be highlighted as drug misuse in Black/African American population is known to be different from White populations.
The following is recommended to support/enrich discussion:
Line 227: Meperidine decreased distribution more likely to be due to change in guidelines than awareness of opioid addiction crisis. (Where has all the meperidine gone? Dobbins E, Nursing2010: January 2010 - Volume 40 - Issue 1 - p 65–66, doi: 10.1097/01.NURSE.0000365924.16631.a4).
Line 253-254: Needs to correlate low methadone use to the absence of NTP in Guam.
The tables are clear however for table 1 it would be interesting to make it easier to visualise opioids change. Maybe different colour for up and down changes (or colours in a scale). Table 2: Missing parenthesis – Hawaii oxycodone)
Figure 1A to C are well designed and easy to understand. But figure not clear what it is representing, is it mean of overtime (2006-2016) MME per person? Better to use scatter plot instead of bar graph.
Figure 2: It would be interesting to have both graphs under the same Y-axis. It would make the graphs easily comparable.

Experimental design

Good experimental design and in the scope of the journal. It clearly fills an identified knowledge gap.
When calculating MME per capita was only 2010 census used? If so, this needs to be added as a limitation to the research. Population numbers can change considerably in 10 years and using the same population to calculate MME per capita over 10 years can introduce an error to this data set.

Validity of the findings

The findings are valid and benefit to literature clearly stated.
Data is robust and raw data can be accessed online.
Line 144: Why paired t-test used to compare the MME per capita by location? It is not clear to me why using paired t-test. Although data is coming from the same database, it is about different sample groups.
Interesting conclusion highlighting answer to the research question and important points which could explain differences found.

Additional comments

This is an interesting manuscript, well written and straight to the point. More information on how some of the differences encountered could be related to specific policies and changes in the territories would have been appreciated.

---

## Round 0.2 · accepted · Accept

The authors have done a good job responding to the comments of the Reviewers, and with the exception of the use of black type of a red background in the response letter, which was hard to take first thing in the morning, it is fine. As an editorial comment (i.e. me editorializing, rather than asking for anything more), I think that it is becoming more reasonable to start citing grey literature, or non-peer reviewed material, as long as it is clear what it is. The speed with which information is becoming available and policy is changing means that standard journal publication is becoming too slow for some things.

#